# The Impact of Critical, Creative, Metacognitive, and Empathic Thinking Skills on High and Low Academic Achievements of Pre-Service Teachers

**DOI:** 10.3390/jintelligence13040050

**Published:** 2025-04-16

**Authors:** Hatice Kumandaş-Öztürk, Özlem Ulu-Kalın

**Affiliations:** Faculty of Education, Artvin Coruh University, 08000 Artvin, Türkiye; ozlemulu@artvin.edu.tr

**Keywords:** creative thinking, critical thinking, metacognitive thinking, empathic thinking, academic achievement, logistic regression

## Abstract

This study aims to determine how higher-order thinking skills—namely creative thinking, critical thinking, metacognitive thinking, and empathic thinking—impact the academic achievement of pre-service teachers at both low and high levels. The study was conducted using the predictive model. The case sampling method, a purposive sampling method, was used in the study. The study participants included 196 volunteer pre-service teachers attending Artvin Coruh University, Faculty of Education. The study data were analyzed using binomial logistic regression analysis. The analysis revealed that the academic achievement of the pre-service teachers varied significantly based on other higher-order thinking skills, except for empathy. Furthermore, the contributions of these variables to academic achievement were ranked based on Exp(β) (odds/likelihood). The findings demonstrated that all variables affected academic achievement, while creative thinking skills contributed most significantly, followed by critical thinking and metacognitive thinking skills. It was also determined that the contribution of empathy skills was not statistically significant (*p* > 0.05). It was observed that the increase in higher-order thinking skills led to greater academic achievement. Similarly, low higher-order thinking skills significantly led to a decrease in achievement. Thus, it could be recommended that learning activities be revised, and the number of activities aimed at improving thinking skills should be increased for the active acquisition of higher-order thinking skills in higher teacher training institutions.

## 1. Introduction

Currently, individuals live in an “information society” where they should possess various qualifications such as investigative skills, skills that allow them to solve complex and novel problems, creative and critical thinking, the ability to organize their thoughts, and skills associated with the recognition and employment of various thinking methods. A qualified and influential individual in this society should possess basic thinking skills ([72]). Basic thinking skills are required for the effective use of knowledge, the development of creative solutions to daily problems and challenges, and socialization. Thinking skills play a key role in personal development at all stages of life, especially in academic settings. Thus, the education system should motivate students and teachers to actively participate in instruction and learning processes to train qualified individuals ([45]).

Thinking, as a cognitive process, is the most crucial part of acquiring knowledge, understanding, and learning. At the same time, thinking serves as the foundation for questioning, evaluating, and generating new knowledge ([42]). Although there are various approaches to classifying levels of thinking, the distinction between lower-order thinking and higher-order thinking is widely accepted in the literature. The fundamental difference in this classification is that higher-order thinking involves using prior knowledge, skills, and experiences gained from the past—through methods such as deduction, induction, and analogy—when solving a problem or engaging in reasoning in cases where available information is insufficient ([64]).

The concept of higher-order thinking is derived from [14]’s ([14]) cognitive taxonomy ([38]). The cognitive domain within this taxonomy encompasses knowledge and the development of intellectual skills ([14]). This taxonomy consists of six main categories, arranged from the simplest to the most complex cognitive processes, which involve recalling or recognizing specific facts, procedural patterns, and concepts. In summary, Bloom classified intellectual behavior into six levels: remembering, understanding, applying, analyzing, evaluating, and creating. While remembering, understanding, and applying are considered lower-order thinking skills as they primarily require basic recognition or recall ([15]), analyzing, evaluating, and creating are classified as higher-order thinking skills ([38]; [64]).

Higher-order thinking skills (HOTS) play a crucial role in enhancing individual and societal well-being as well as the quality of democratic structures ([113]). Technological advancements and the increasing significance of artificial intelligence are affecting many professions and fields. However, AI technologies have not yet surpassed individuals who possess higher-order thinking competencies such as judgment, decision-making, critical thinking, and creative thinking. In this regard, teachers need to possess these competencies to cultivate individuals with these skills. Approaches that place the student at the center of the learning experience, encourage thinking, utilize Socratic questioning, adopt inquiry-based learning, incorporate project-based and experiential methods, and promote collaboration are effective in fostering HOTS in individuals ([98]; [57]). Additionally, instructional methods and activities such as performance tasks, discussions, drama, real-world activities, and case studies significantly contribute to the development of higher-order thinking skills (HOTS) in higher education and teacher training institutions ([12]).

The framework of basic thinking skills is employed to describe higher-order thinking skills and is used as a general term to describe thinking processes ([75]). Higher-order thinking skills have also been described as the capacity to go beyond available knowledge to classify, infer, generalize, and solve problems in complex situations ([67]). Higher-order thinking skills are required when encountering unfamiliar problems, uncertainties, questions, or dilemmas ([70]).

Higher-order thinking skills include several skills that are required in case of challenges that cannot be solved immediately, force the individual to try several methods to solve, require a different thinking approach, and allow the individuals to get to know themselves. Furthermore, the following have been described as traits that do not reflect higher-order thinking skills ([115]):Unquestioning acceptance of data input;Non-questioning;Inability to determine the common denominator across current and prior knowledge;Acceptance of previous methods instead of trying novel solutions.

Higher-order thinking skills require abstract thinking, the ability to imagine various dimensions of objects, categorize, and associate them with other concepts, distinguish the similarities and the differences, and combine these with holistic experiences and present them based on mental processes ([20]).

Higher-order thinking skills have also been described as the ability to detail a given material, render it understandable, infer from the known, and associate ([90]). Individuals with these skills not only require a strong vocabulary, syntax, and numerical skills but also should know how to express their experiences and ideas. Thus, individuals who reach the analysis and creation stages of higher-order thinking based on Bloom’s taxonomy would be more successful and could actively solve unpredicted economic and social solutions ([46]).

The views on thinking skills that are included in higher-order thinking vary. However, common cognitive skills included by authors are creative thinking, critical thinking, and reflective thinking ([60]). [70] ([70]) included metacognitive thinking skills among higher-order thinking skills, along with creative and critical thinking. Empathy has also been considered a cognitive skill since the 1950s; however, it is currently considered an affective skill ([109]).

Emphasized higher-order thinking skills include creative thinking and critical thinking skills. Creative thinking is the ability to develop novel ideas, solutions, or products in emerging events or situations. It entails the ability of an individual to develop unique and original solutions to daily life problems. Common creative skill traits include the ability to associate previously unassociated objects with concepts, events, and situations, the ability to acquire different experiences, and the ability to try unique methods ([104]).

Creative thinking skills include several skills. Significant skills include sensitivity to problems, the development of opinions and associations, flexibility, originality, elaboration, and re-description ([79]). Although creativity cannot be developed within a short time, it can be instructed and improved under adequate conditions. The development of creative thinking skills depends on the availability of a non-stressful and peaceful environment where the individual can feel free and safe ([81]). Creative thinking skills include unordinary thinking, flexibility, fluency, and originality under different conditions.

In addition to creative thinking, critical thinking skills are also included among higher-order thinking skills. Critical thinking is described as the ability to analyze facts, develop and organize ideas, defend opinions, make comparisons, draw conclusions, analyze discussions, and solve problems ([48]; [35]).

The primary critical thinking acts include “investigation, comprehensive thinking, free-thinking, and reconstruction.” Thus, the individual can perform the following actions ([74]):Query, analysis, evaluation, and discussion of assumptions during investigation.Investigation of the causality of decisions during comprehensive thinking.Independence during free thinking.Confirming, correcting, or replacing the existing value system with a new belief during reconstruction.

The employment of critical thinking skills begins at birth and continues throughout life ([66]). Because individuals observe all the events around them starting from infancy and make certain inferences, based on these inferences, they begin to establish causality ([110]). Later, as children complete cognitive development, they start to perform this skill consciously. Thus, instruction on critical thinking skill sub-dimensions from an early age allows children to think multidimensionally because children with critical thinking skills are like scientists who are curious about discovering everything ([37]). Also, the acquisition of these skills at an early age determines the individual’s approach to problem solving in later years and contributes to the academic development of the individual ([59]).

Metacognitive thinking skills are also considered a subset of the higher-order thinking skills. The concept of metacognition is significant in learning since it is assumed that metacognition supports learner autonomy ([117]). The concept of metacognition, first introduced by [36] ([36]), was described as an individual’s awareness of their personal learning process. Thus, the individuals’ ability to know how they learn best, develop and use effective learning strategies, and make self-assessments about what and how much they learned are the outcomes of metacognitive thinking skills.

Previous studies have reported that planning, monitoring, and analysis, which help individuals organize their cognitive processes in the learning environment, can improve metacognitive awareness and thus allow the control and self-regulation of cognitive thinking, learning, and products ([47]; [119]). According to [63] ([63]), the instruction of metacognitive strategies such as planning, monitoring, and analysis can contribute to lifelong reflective thinking, problem solving, responsibility, and self-confidence in rapid decision-making. Metacognitive strategies that include planning, monitoring, and analysis can improve the metacognitive awareness of students about cognitive resources and the effective employment of these resources. Successful learning can be achieved with metacognitive awareness ([68]).

Similarly to creative thinking, critical thinking, and metacognitive thinking skills, empathy skills are also included among higher-order cognitive thinking skills. Empathy was described by German psychologist Lipps as individuals’ knowledge of objects, themselves, and other individuals during communication ([44]).

Empathy has been described as follows in the literature:A response similar to the emotional and cognitive state of another individual ([88]).An individual’s efforts to accurately understand the emotions and thoughts of another individual ([30]).Recognition of the emotions of others and the factors that led to these emotions ([53]).Recognition and management of one’s own emotions, as well as sensitivity to the emotions, desires, and needs of others ([41]).The ability to put oneself in someone else’s shoes regarding emotions and thoughts ([103]).

Empathy entails responses to emotions, sharing what others feel, and reflection on these emotions like a mirror, and it allows the establishment of good relations with others, understanding others, and sharing their emotions ([61]). Empathy is also a fundamental human skill that regulates relationships and promotes cooperation and group harmony ([91]). Empathy skills demonstrate that not everyone shares the same ideas, and individuals can have different views, emotions, and ideas ([29]).

Students who are educated in an empathic school environment feel safe, loved, and socially confident ([65]), because in an empathic classroom, students can freely express their views ([82]). An empathic teacher can help improve student self-esteem and contribute not only to cognitive development but also to student personality ([51]).

Higher-order thinking skills are generally discussed separately in the literature. Certain studies have addressed only a single skill, while others have been conducted on several skills and the correlations between these skills. Furthermore, certain studies have investigated the correlations between these skills and academic achievement. Certain studies on the correlation between creative thinking and academic achievement have reported a positive correlation ([2]; [33]), while others have reported an insignificant correlation ([55]; [8]). Certain studies have reported a positive significant correlation between critical thinking skills and metacognitive thinking ([6]; [94]). Others have reported a significantly weak correlation between metacognitive thinking skills and academic achievement ([52]; [10]; [19]; [93]; [107]), while some have reported no correlation between academic achievement and metacognitive thinking skills ([34]; [95]).

Higher-order thinking skills are not merely the ability to recall information but also the competencies that enable individuals to confront challenges, difficulties, questions, or dilemmas encountered in both education and daily life. Moreover, these skills—especially critical thinking, creative thinking, metacognitive thinking, and similar cognitive abilities—are utilized to predict students’ academic achievement ([102]). Therefore, it is crucial to introduce individuals to assessment tools based on higher-order thinking skills that prepare them to solve new problems, adapt to new environments, and make decisions on specific issues ([89]).

Thinking skills are also closely related to learning. In this context, an individual’s thought process can influence their learning capacity, speed, and performance. Improving their thinking abilities positively may impact their learning development. Students with higher-order thinking skills possess the ability to learn effectively, perform well, and minimize deficiencies. When individuals encounter problems requiring lower-order thinking skills, their cognitive capacities diminish, negatively affecting the development of higher-order thinking skills ([102]). Conversely, when individuals frequently engage in situations demanding higher-order thinking skills, they engage in more mental activity. This, in turn, compels individuals to be more creative ([112]). From this perspective, examining the impact of higher-order thinking skills on academic achievement is significant. This study investigates how higher-order thinking skills—namely, critical thinking, creative thinking, metacognitive thinking, and empathic thinking—affect the academic achievement (low and high levels) of pre-service teachers.

The review of studies on higher-order thinking skills and academic achievement revealed no survey of the association between these skills and academic achievement. Furthermore, there is no study on the correlations between academic achievement and these skills or the impact of these skills on achievement. Moreover, it is important for teacher candidates to have higher-order thinking skills and to determine the relationship between academic achievement and thinking skills to fill the gap in the literature. Also, in Turkey, the Ministry of National Education revised the curricula at all educational levels in 2024, and the acquisition of higher-order thinking skills was accepted as the main theme in all programs ([73]). Thus, it is important to determine the relationship between these skills and the academic achievement of pre-service teachers, as this would affect their graduation and appointment. Conducting this study based on these requirements, issues, and the description of the status according to the study findings could lead to the elucidation of new perspectives for future researchers. Thus, the present study aimed to determine how higher-order thinking skills, such as creative thinking, critical thinking, metacognitive thinking, and empathy, affect the academic achievement (low and high level) of pre-service teachers.

The hypothesis of this study is as follows: the higher-order thinking skills (HOTS) of creative thinking, critical thinking, metacognitive thinking, and empathy influence the academic success levels (high or low) of pre-service teachers.

## 2. Materials and Methods

The first step of a scientific study is research design and determining methodology. One way to design research is to use the data collection process suggested by the “research onion” model developed by [96] ([96]). Initially, the research model was selected; then, the study group and the data collection method were determined.

### 2.1. The Research Model

In this study, a predictive model was utilized to identify variables that significantly predict a specific outcome or criterion ([39]; [25]). More specifically, the predictive model was employed to analyze the factors influencing academic achievement levels.

### 2.2. The Study Groups

The study sample was assigned with the typical case sampling method, a purposive sampling approach. Purposive sampling allows the selection of information-rich cases that are not based on probability ([23]) and the conduction of an in-depth investigation. On the other hand, typical case sampling includes ordinary and average cases within society ([86]). Thus, typical cases are used to represent the population based on similar characteristics ([71]). Therefore, the study data were collected from 196 volunteer pre-service teachers who attended Artvin Coruh University, Faculty of Education. This public university accepts students based on scores from the Higher Education Institutions Exam, a national exam in Turkey. Participant demographics are presented in Table 1.

### 2.3. Data Collection Instruments

Since the present study was conducted on higher-order thinking skills such as critical, empathic, creative, and metacognitive thinking skills, instruments developed to measure these skills were employed. The measurement tools and data collection methods were approved by the Artvin Coruh University Ethics Committee on 15 December 2023 (Approval no: E-114974). The measurement instruments employed in the study are presented below.

Demographic data form: The form was developed by the authors and included demographic questions (gender and department of the pre-service teachers).Academic achievement: In this study, academic achievement, which serves as the dependent variable, is defined as the overall academic grade point average (GPA) of pre-service teachers. Data on this variable were collected through a short-response question in the “Demographic data form”. Academic GPA was self-reported by students.Creative thinking aptitude scale: Marmara’s creative thinking attributes scale, developed by [83] ([83]), includes 25 items and six factors. Cronbach’s alpha internal consistency coefficient is 0.93 for the entire scale.Critical thinking aptitude scale: The scale was adapted to the Turkish language by [56] ([56]) and includes 25 items. The Cronbach alpha internal consistency coefficient was calculated as 0.92 with the data collected in the present study for the entire scale, 0.87 for the participation dimension, 0.79 for the cognitive maturity dimension, and 0.78 for the innovation dimension.Metacognitive thinking scale: The scale was adapted to the Turkish language by [18] ([18]). In the present study, 12 items in the metacognitive thinking subscale of the main scale were employed. The scale is a seven-point Likert-type scale. In the present study, the Cronbach alpha coefficient of the scale was calculated as 0.84.Basic empathy scale: The Likert-type scale was adapted to the Turkish language by [109] ([109]). In the present study, the Cronbach alpha coefficient of the scale was calculated as 0.81 for the emotion dimension and 0.78 for the cognitive dimension.

The construct validity of the scales employed in the study was analyzed with confirmatory factor analysis and the LISREL 8.8 software. In the confirmatory factor analysis, the χ^2^/df, RMSEA (Root Mean Square Error of Approximation), AGFI (Adjustment Goodness of Fit Index), NNFI (Non-Normed Fit Index), and CFI (Comparative Fit Index) were calculated to determine the statistical fit criteria and verification of the model. For a good model fit, the χ^2^/df (degrees of freedom) ratio should be between 2 and 3 ([118]). The χ^2^/df ratio and RMSEA findings were as follows:Creative thinking scale: χ^2^/df = (350.87/275) = 1.27, RMSEA = 0.08.Critical thinking scale: χ^2^/df = (322.42/275) = 1.17; RMSEA = 0.07.Metacognitive thinking scale: χ^2^/df = (60.03/54) = 1.11; RMSEA = 0.06.Basic empathy scale: χ^2^/df = (190.67/169) = 1.12; RMSEA = 0.05.

These findings indicated that the fit of the scales and the data was acceptable, and construct validity was confirmed ([16]). A figure less than or equal to 0.05 indicates a perfect fit, while a figure greater than 0.10 indicates an unfit model ([118]). In the present study, the comparative fit indices, the CFI, AGFI, and NNFI, were between acceptable limits ([26]) for all three measurement instruments (CFI > 0.88, AGFI > 0.90 and NNFI > 0.95).

### 2.4. Data Analysis

Binomial logistic regression analysis was employed to determine the impact of high-level empathy, critical thinking, creative thinking, and metacognitive thinking skills of pre-service teachers on their academic achievement levels. Binomial logistic regression analysis was employed to estimate the impact of independent variables on dependent variables, where the dependent variables included binomial or multiple categorical data ([1]). Logistic regression analysis does not require the confirmation of assumptions such as the normal distribution of independent variables, linearity, or equality of variance–covariance matrices and is preferred when the dependent variable is categorical (present–absent, successful–unsuccessful, etc.) ([101]). This analysis method determines factors (independent variables) that are expected to affect the probability of one of the categories. In logistic regression, the impact of independent variables on the dependent variable is determined as probabilities ([80]).

In the logistic regression analysis employed in the current study, the academic achievement of the pre-service teachers (successful–unsuccessful) was determined as the dependent variable, and higher-order thinking skills were determined as independent variables.

The Kolmogorov–Smirnov normality test was applied to determine whether the dependent variable, the academic achievement scores of the pre-service teachers, exhibited normal distribution. The analysis revealed that academic achievement data were not distributed normally. This finding is presented in Table 2.

Since the dependent variable, academic achievement, did not exhibit a normal distribution, a logistic regression analysis, which does not require the assumption of normality, was conducted. In the analysis, the academic achievement score was coded based on the CC grade, accepted as the cut-off value for passing the course and graduation. Thus, the CC score, the cut-off value for passing the course, and higher scores were classified as high achievement (1), while scores below the cut-off value were classified as low achievement (0). Analyses were carried out based on the cut-off value.

Before the interpretation of the logistic regression analysis findings, assumptions (multicollinearity, outliers, and model data fit) were tested. In the multicollinearity between independent variables, it was determined that the tolerance was between 0.56 and 0.90 and the variance inflation factor (VIF) was between 1.11 and 2.35, while the highest condition index (CI) was 26.65. These findings demonstrated that there was no significant multicollinearity between independent variables. Standardized residuals were investigated to determine outliers, and errors were within the accepted range of ±3, demonstrating no outliers. The data were analyzed with IBM SPSS v.27 (Statistical Package of the Social Sciences) software.

Regression coefficients, Wald statistics, degrees of freedom, standard error, 95% confidence intervals, odds (likelihood) ratios, and correct classification thresholds for independent variables were calculated in the logistic regression analysis. In all analyses, the statistical significance level was accepted as 0.05.

In this study, the effect of independent variables on the dependent variable was examined only through logistic regression analysis. The fact that potential interaction/moderation variables that could influence this effect were not considered or that statistical analyses regarding possible mediation relationships were not conducted constitutes the limitation of this study.

## 3. Results

The dependent variable in this study was examined in two categories. Accordingly, pre-service teachers classified as having low academic achievement constituted 30.1% (N = 59) of the participants, while those with high academic achievement accounted for 69.9% (N = 137). Descriptive statistics regarding independent variables are presented in Table 3.

Four independent variables were included in the logistic regression analysis conducted to determine the variables that predicted the probability of academic achievement of pre-service teachers, and these variables contributed significantly to the probability equation formulated to predict the probability of academic achievement. The logistic model and the data fit were tested with the Hosmer–Lemeshow test. The Hosmer–Lemeshow test demonstrated a good model fit (χ^2^_(8)_ = 3.88 *p* > .01). The β parameters determined in the logistic regression analysis, Wald statistics, degrees of freedom (df), significance levels (sig.), standard error (St.Er.), and Exp(β) (odds/likelihood) for these parameters are presented in Table 4.

As seen in Table 4, the academic achievements of pre-service teachers significantly varied based on the creative thinking, critical thinking, and metacognitive thinking variables, demonstrating that these variables had a significant impact on the possibility of reading comprehension. Empathy skills did affect academic achievement, albeit insignificantly. The ranking of the impact of each variable on academic achievement based on Exp(β)/likelihood revealed the following findings:The academic achievement of pre-service teachers increased by 1.109 with the increase in their creative thinking score (*p* < 0.05).The academic achievement of pre-service teachers increased by 1.106 with the increase in their critical thinking score (*p* < 0.05).The academic achievement of pre-service teachers increased by 1.041 with the increase in their metacognitive thinking score (*p* < 0.05).The increase in empathy skills positively affected academic achievement; however, the effect was not significant (*p* > 0.05).The above-mentioned findings demonstrated that the impact of creative thinking skills was the most significant on academic achievement, while empathy skills had the least effect.

The classification success of the logistic model developed for the academic achievements of pre-service teachers is presented in Table 5.

As seen in Table 5, the accurate classification rate of the observations by the logistic regression model was 86.2%. Thus, the model assigned 129 pre-service teachers with high academic achievement and 49 pre-service teachers with low academic achievement into the accurate group.

The scatter diagram of the independent study variables based on the dependent variable, academic achievement, is presented Figure 1.

Since Figure 1 is a scatter diagram of the variables of the study, it also helps to visually interpret the findings regarding the classification of variables.

In this study, the Cox and Snell R^2^ value of the model was found to be 0.54. Together, these variables explain 54% of the total variance in academic achievement scores.

[22] ([22]) proposed the formula f^2^ = (R^2^/(1 − R^2^)) to calculate the standardized effect size in all regression analyses. Based on this value, the effect size was computed as f^2^ = (0.54/0.46) = 1.17. This result indicates that the effect size of the variables is considerably large.

## 4. Discussion

This section presents the study findings and links them to those reported in the literature. Then, certain recommendations are included.

In the present study, conducted with the data collected from pre-service teachers and the predictive model, it was determined that the academic achievements of pre-service teachers varied significantly based on high-level thinking skills except for empathy. Metacognitive thinking played a key role in both the personal and professional development of pre-service teachers. When pre-service teachers could manage the learning process and employ effective strategies, they could achieve higher academic achievement. The development of these skills is accepted as a significant step towards becoming an effective teacher.

In the study, the ranking of the contributions of creative thinking, critical thinking and empathy variables to academic achievement was investigated based on Exp(β)/likelihood. The findings demonstrated that although all variables had an effect on academic achievement, the impact of creative thinking skills was the highest. Certain previous reports are consistent with this finding ([55]; [8]; [17]; [32]; [2]; [33]).

The study also determined that critical thinking and metacognitive thinking contributed to academic achievement in that order. Several reports in the literature are consistent with this finding ([3]; [5]; [4]; [9]; [11]; [13]; [21]; [24]; [58]; [62]; [69]; [78]; [97]).

Finally, it was determined that the correlation between empathy and academic achievement was insignificant. Individuals with empathy had better sensitivity, cooperation, sharing and collaboration skills ([85]; [92]). In a study conducted by [77] ([77]), a positive correlation was found between students’ empathic behaviors and their academic performance. However, [105] ([105]) reported that cognitive empathy led to lower participation in students’ learning activities. Similarly, [49] ([49]) found no relationship between empathy and academic performance. In another study, [111] ([111]) examined the relationship between empathy skills, emotional intelligence levels, and parental attitudes among gifted and normally developing elementary school students. The study concluded that empathy skills increased among normal children raised in democratic family environments, whereas empathy skills decreased among gifted children. This finding is particularly noteworthy.

Overall, the results of this study suggest that developing higher-order thinking skills plays a crucial role in enhancing academic achievement. Higher-order thinking skills involve applying learned knowledge to real-life situations. In this context, school success is a key factor in utilizing these skills. A student with well-developed higher-order thinking skills can synthesize existing and newly acquired information to develop alternative solutions to a given problem ([54]). Among the core components of higher-order thinking skills (HOTS), critical thinking and creative thinking hold particular significance. Additionally, the ability to organize one’s thinking is also a fundamental aspect of HOTS ([114]; [75]). Therefore, individuals with well-developed metacognitive thinking skills are likely to have enhanced abilities in other cognitive domains as well. The literature highlights that the development of higher-order thinking skills enhances individuals’ learning speed and abilities. In other words, students with advanced thinking skills can learn more efficiently, address their shortcomings, and improve their performance ([116]). Investigating the impact of these variables on academic achievement provides valuable insights into student development.

Based on the present study findings, the development of creative thinking skills could help improve academic achievement. According to [100] ([100]), “there is no uncreative individual. There are only individuals who are more or less locked, blocked, frozen and in need of long- or short-term education”. Thus, individuals’ creative skills could be activated and improved with creativity education. Previous studies reported that creativity could be developed through education. Various education programs have been tested, including a creativity education program conducted by Torrance and Safter for higher education students. It was concluded that implementing this program could improve creativity, which was influenced by socio-psychological and cognitive variables ([7]). Furthermore, activities available in most courses that can be enhanced with pictures, shapes, and diagrams, as well as small jokes and dialogs that entertain and motivate the student, are significant for the achievement of permanent and effective learning.

The study findings demonstrated that critical thinking and metacognitive thinking also contribute to academic achievement. Thus, courses could be included in college curricula for the acquisition or development of critical thinking and metacognitive thinking skills. This finding is consistent with reports in the literature. [50] ([50]), [27] ([27]), [40] ([40]), and [99] ([99]) reported that there was a positive correlation between academic achievement and critical thinking. [108] ([108]) reported that both critical thinking and metacognitive skills contributed to problem-solving skills. In the study, it was also determined that there was a positive correlation between these two skills ([6]). Furthermore, [87] ([87]) reported that critical thinking skills improve in learning environments where questioning learning approaches are adopted.

It was also determined, in the present study, that empathy did not contribute significantly to academic achievement. Previous studies classified empathy as an innate ability; however, it is currently considered a skill that can be learned and instructed ([84]). Certain authors consider empathy an innate ability, while others consider it a skill acquired through experiences. The currently accepted view considers empathy an innate ability that can also be instructed ([43]) and is only one of the functions of the brain ([28]). Thus, although empathy did not contribute significantly to academic achievement, it is certain that empathy is an important skill that pre-service teachers should acquire. An empathic teacher can understand their students and help them acquire empathy during communication ([31]). Therefore, applications that aim for the acquisition of higher-order thinking skills should be included in learning activities in all educational institutions, especially teacher training schools. The following can be recommended based on the study findings. Based on the present study findings, the number of instructional methods and course activities that can develop thinking skills should be increased in tertiary education and course content, improving academic achievement. In curricula published by the Ministry of National Education on the new preschool, first grade and fifth grade of primary school, high school preparatory, and ninth grade of high school instructional models for the 2024–2025 academic year ([73]), high-level thinking skills were particularly emphasized and related activities were frequently noted. Thus, future studies should be conducted to determine and improve the skills of pre-service teachers. [106] ([106]) emphasized the significance of teachers’ thinking skills in supporting students’ development and enhancing the quality of education. Activating higher-order thinking skills effectively improves classroom achievement and leads to desired learning outcomes ([76]). Thus, teachers’ active use of these skills would increase the quality of education.

The fact that the study data were collected only from students attending the Faculty of Education at a single university was a limitation of the study. Studies conducted with a larger, randomly selected group would lead to more generalizable findings. Thus, future studies should be conducted with larger groups, utilizing these variables. Certain higher-order thinking skills were not addressed in the current study. Thus, it could be suggested that future studies should address higher-order thinking skills based on various variables associated with academic achievement. Furthermore, based on the findings of the present study, a qualitative dimension can be included regarding the development of these skills and their contributions to pre-service teachers in collecting more in-depth data.

## Figures and Tables

**Figure 1 jintelligence-13-00050-f001:**
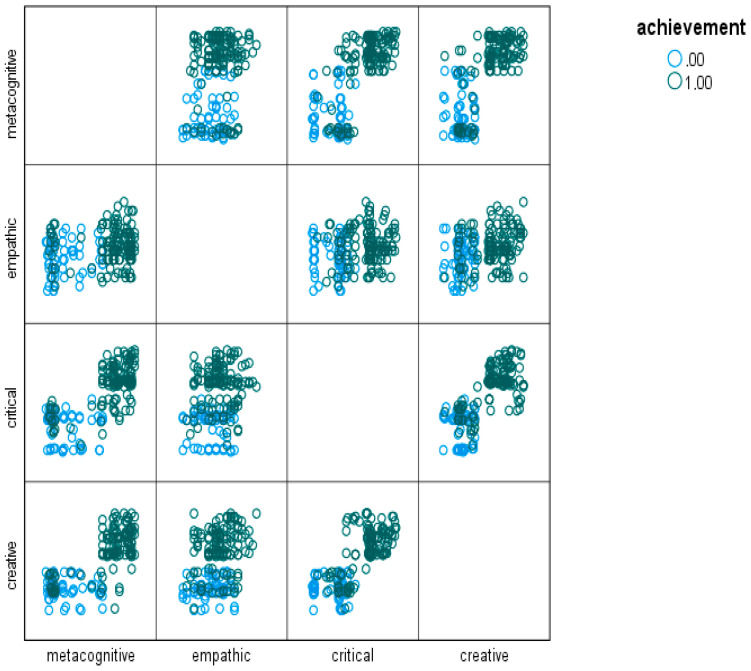
The distribution of the independent study variables based on the dependent variable.

**Table 1 jintelligence-13-00050-t001:** Participant demographics.

Variable		N	%
Department	Mathematics Instruction	30	15.3
	Preschool Instruction	29	14.8
	Guidance–Psychological Counseling	15	7.7
	Primary School Instruction	24	12.2
	Turkish Language Instruction	44	22.4
	Social Studies Instruction	54	27.6
Gender	Female	126	64.3
	Male	70	35.7

**Table 2 jintelligence-13-00050-t002:** Normality test results.

Academic achievement	**Kolmogorov–Smirnov Statistic**	**df**	**Sig.**
	0.141	196	0.000

**Table 3 jintelligence-13-00050-t003:** Descriptive statistics regarding independent variables.

İndependent Variables	N	Minimum	Maximum	Mean	Std. Deviation
Empathic	196	52	72	61.28	4.09
Critical	196	54	118	88.35	16.88
Creative	196	64	125	95.20	16.39
Metacognitive	196	21	79	54.82	20.00

**Table 4 jintelligence-13-00050-t004:** Logistic regression model and significance of the variables.

Variable	β	St.Er.	Wald	df	Sig.	Exp(β)
Constant	−22.355	5.390	17.200	1.000	0.000 *	0.000
Creative thinking	0.104	0.038	7.577	1.000	0.006 *	1.109
Critical thinking	0.100	0.033	9.359	1.000	0.002 *	1.106
Metacognitive thinking	0.040	0.018	5.145	1.000	0.023 *	1.041
Empathy	0.076	0.068	1.247	1.000	0.264	1.079

* *p* < 0.05.

**Table 5 jintelligence-13-00050-t005:** Classification success of the logistic regression model.

Observed Achievement			Predicted Achievement		Accuracy Percentage
Step 1	Achievement	0.00	49	10	83.1
		1.00	17	129	87.6
	Overall Percentage		0		86.2

## Data Availability

The data presented in this study are available on request from the corresponding author to protect the confidentiality of the participants.

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
