# Peer review of "The Impact of Critical, Creative, Metacognitive, and Empathic Thinking Skills on High and Low Academic Achievements of Pre-Service Teachers"

_jintelligence, 2025, doi:10.3390/jintelligence13040050_

Round 1
Reviewer 1 Report
Comments and Suggestions for Authors
1. The introduction does not adequately establish the research problem or justify its significance. The authors should explicitly state the gap in the literature and formulate clear research questions and hypotheses. Additionally, the discussion of higher-order thinking skills should be expanded to include recent theoretical advancements in cognitive and educational psychology.
2. The rationale for using a relational survey model is unclear. The authors should justify why this design was chosen over experimental or longitudinal approaches and clarify how it effectively addresses the research objectives.
3. While validated scales were used, there is insufficient discussion of their reliability and validity in this study’s context. The authors may consider perform confirmatory factor analysis (CFA) to ensure construct validity and report Cronbach’s alpha values for reliability.
4. The binary classification of academic achievement using a 2.70 GPA threshold is arbitrary and should be justified with references or alternative statistical reasoning. Additionally, the logistic regression model lacks explanation regarding assumptions, effect sizes, and multicollinearity diagnostics. A more comprehensive statistical approach, such as multiple regression or structural equation modeling (SEM), should be considered.
5. The results section lacks depth and does not provide adequate explanations for the findings. The authors should report descriptive statistics (means, standard deviations) and effect sizes and explore potential confounding variables that may have influenced the results. Further analysis, such as interaction effects or moderating relationships, could strengthen the study’s contributions.
6. The discussion does not meaningfully connect findings to prior research or theoretical frameworks. The authors should compare their results with existing literature, discuss the broader implications for teacher education, and explore possible explanations for why empathy did not significantly impact academic achievement. Additionally, limitations should be explicitly acknowledged, and future research directions should be suggested.
Author Response
Response
Reviewer#1:
Thank you for all your suggestions. You can find all edits in the file I uploaded to the system in red. Kind regards.
1. The introduction does not adequately establish the research problem or justify its significance. The authors should explicitly state the gap in the literature and formulate clear research questions and hypotheses. Additionally, the discussion of higher-order thinking skills should be expanded to include recent theoretical advancements in cognitive and educational psychology.
The research problem revised. You can find all edits in the file revised uploaded to the system in red
2 The rationale for using a relational survey model is unclear. The authors should justify why this design was chosen over experimental or longitudinal approaches and clarify how it effectively addresses the research objectives.
This section has been revised.
3. While validated scales were used, there is insufficient discussion of their reliability and validity in this study’s context. The authors may consider perform confirmatory factor analysis (CFA) to ensure construct validity and report Cronbach’s alpha values for reliability.
This information is already included between the lines Page 6 L255-L288).
- The binary classification of academic achievement using a 2.70 GPA threshold is arbitrary and should be justified with references or alternative statistical reasoning. Additionally, the logistic regression model lacks explanation regarding assumptions, effect sizes, and multicollinearity diagnostics. A more comprehensive statistical approach, such as multiple regression or structural equation modeling (SEM), should be considered.
The justification for the binary classification of academic success using the 2.70 GPA threshold is given. This analysis was used for the purpose of the study. New research can be done for other purposes and used in the analyses you suggest. Thank you for your information on this point.
. The results section lacks depth and does not provide adequate explanations for the findings. The authors should report descriptive statistics (means, standard deviations) and effect sizes and explore potential confounding variables that may have influenced the results. Further analysis, such as interaction effects or moderating relationships, could strengthen the study’s contributions.
Revised. Descriptive statistics were included, but due to the nature of the analyses, effect size values ​​could not be provided.You can find all edits in the revize file uploaded to the system in red.
- The discussion does not meaningfully connect findings to prior research or theoretical frameworks. The authors should compare their results with existing literature, discuss the broader implications for teacher education, and explore possible explanations for why empathy did not significantly impact academic achievement. Additionally, limitations should be explicitly acknowledged, and future research directions should be suggested.
This section revised.

Reviewer 2 Report
Comments and Suggestions for Authors
This is a valuable paper, primarily due to the significance of the issues it addresses. The topic is timely and aligns well with contemporary discourses in educational research. Studies of this kind can significantly expand existing knowledge regarding the factors responsible for the outcomes of higher education. The decision to explore a topic that has been largely overlooked in previous research was a commendable choice. Indeed, the project aimed to fill a gap in the existing knowledge on the relationship between cognitive skills and the outcomes of initial teacher education.
The value of the article could be further enhanced by making several editorial revisions. First, the research project should be presented with greater precision. In the methodological section, there is no explicit presentation or justification of the research problems or hypotheses, which is a necessary requirement in correlational studies. Additionally, the paper lacks operational definitions of the research variables—particularly the dependent variable.
Another important issue is the absence of a discussion on the conceptualization of key analytical categories in the study. Terms such as creative thinking, critical thinking, metacognitive thinking, and empathic thinking are assigned different meanings depending on the psychological theories underlying them. The reader may expects a clear statement regarding the theoretical framework adopted in the study. Arbitrary or dictionary-based definitions alone are insufficient in this context.
Furthermore, the conceptualization and operationalization of academic achievement remain unclear.
I also suggest that the discussion section emphasize potential directions and methodological solutions for future research on this highly important issue.
I recommend the publication of this paper after addressing the aforementioned concerns and making the necessary revisions.
Author Response
Reviewer 2: response
Thank you for all your suggestions. You can find all edits in the file I uploaded to the system in red. Kind regards.
This is a valuable paper, primarily due to the significance of the issues it addresses. The topic is timely and aligns well with contemporary discourses in educational research. Studies of this kind can significantly expand existing knowledge regarding the factors responsible for the outcomes of higher education. The decision to explore a topic that has been largely overlooked in previous research was a commendable choice. Indeed, the project aimed to fill a gap in the existing knowledge on the relationship between cognitive skills and the outcomes of initial teacher education.
The rationale for the research problems was revised. In addition, the operational definitions of the research variables, especially the dependent variable, were revised in the article.
The value of the article could be further enhanced by making several editorial revisions. First, the research project should be presented with greater precision. In the methodological section, there is no explicit presentation or justification of the research problems or hypotheses, which is a necessary requirement in correlational studies. Additionally, the paper lacks operational definitions of the research variables—particularly the dependent variable.
Another important issue is the absence of a discussion on the conceptualization of key analytical categories in the study. Terms such as creative thinking, critical thinking, metacognitive thinking, and empathic thinking are assigned different meanings depending on the psychological theories underlying them. The reader may expects a clear statement regarding the theoretical framework adopted in the study. Arbitrary or dictionary-based definitions alone are insufficient in this context.
Furthermore, the conceptualization and operationalization of academic achievement remain unclear. I also suggest that the discussion section emphasize potential directions and methodological solutions for future research on this highly important issue. I recommend the publication of this paper after addressing the aforementioned concerns and making the necessary revisions.
The theoretical framework has been revised, an operational definition of academic success has been made and information has been added. In addition, the results and discussion sections have been revised in line with your suggestions.

Reviewer 3 Report
Comments and Suggestions for Authors
The issue addressed in the manuscript is relevant. However, some issues deserve attention and should be improved.
The structure is according to the standards, and it is based on an extensive list of references, even though there should be more references from the last 5 years.
The title and the research objective (see abstract and end of section 1) announce an analysis of the impact on two groups formed based on their levels of achievements, but in the end, a comparison of the two achievement-based groups is not presented. This means that the title and the objective, analysis and conclusion must be better aligned
The overall structure of the abstract is clear, but it will need to be reviewed due to the previous comment.
The introduction covers the main relevant topics and issues, even though some more recent references should be included, namely on research that relates academic achievement to higher-order thinking
The target group section should provide more information on participants’ academic achievement levels (as data on achievement were not collected), even though different levels of achievement were not considered.
Information on higher order thinking skills data collection instrument is well presented. No characterization of the variable achievement makes it hard to understand the research and analysis carried out. There is only a mention of it in brackets (in terms of successful-unsuccessful) in the data analysis section, but this is not explained and is not considered in the conclusion.
The results section is too technical. Additional information/explanation to make it understandable to non-experts in the statistical analysis performed is required. The same applies to Figure 1.
The discussion is interesting, but it should be better aligned with the objective of the study in that it should explicitly address and compare the impact of the different kinds of HOT in each of the academic levels considered. Is HOT more important for/ favorable to low or high achievers?
A conclusion with an explicit answer to the research problem should be included.
Hope thsi comments help you to imporve the manuscript.
Comments on the Quality of English LanguageThe quality of english is good.
Author Response
Reviewer 3: response
Thank you for all your suggestions. You can find all edits in the file I uploaded to the system in red. Kind regards.
The title and the research objective (see abstract and end of section 1) announce an analysis of the impact on two groups formed based on their levels of achievements, but in the end, a comparison of the two achievement-based groups is not presented. This means that the title and the objective, analysis and conclusion must be better aligned. The overall structure of the abstract is clear, but it will need to be reviewed due to the previous comment.
The result sections have been revised in terms of comparisons regarding low and high academic achievement.
The introduction covers the main relevant topics and issues, even though some more recent references should be included, namely on research that relates academic achievement to higher-order thinking. The target group section should provide more information on participants’ academic achievement levels (as data on achievement were not collected), even though different levels of achievement were not considered.
The academic achievement variable was defined and necessary explanations were made.
Information on higher order thinking skills data collection instrument is well presented. No characterization of the variable achievement makes it hard to understand the research and analysis carried out. There is only a mention of it in brackets (in terms of successful-unsuccessful) in the data analysis section, but this is not explained and is not considered in the conclusion.
Revised. You can find all edits in the file revised uploaded to the system in red
The results section is too technical. Additional information/explanation to make it understandable to non-experts in the statistical analysis performed is required. The same applies to Figure 1.
Revised. You can find all edits in the file revised uploaded to the system in red
The discussion is interesting, but it should be better aligned with the objective of the study in that it should explicitly address and compare the impact of the different kinds of HOT in each of the academic levels considered. Is HOT more important for/ favorable to low or high achievers?
A conclusion with an explicit answer to the research problem should be included.
The points you mentioned - especially the effect of higher-order thinking skills on low and high academic success - have been commented more functionally and understandably.
Thank you

Round 2
Reviewer 1 Report
Comments and Suggestions for Authors
The manuscript has been significantly improved, but the following minor revisions are still recommended:
-
Explicitly state the hypothesis rather than implying it in the research question.
-
Consider adding interaction/moderation analyses if feasible. If not, acknowledge this as a limitation.
-
Expand the introduction slightly to incorporate more recent theoretical advancements in cognitive and educational psychology.
Author Response
Thank you for all your suggestions. You can find all edits in the revised file uploaded to the system in green. Kind regards.
- Explicitly state the hypothesis rather than implying it in the research question.
Hypothesis added. You can find it on page 5 (lines 236-239) in the file I uploaded to the system
- Consider adding interaction/moderation analyses if feasible. If not, acknowledge this as a limitation.
This suggestion was added as a limitation on page 8 (lines 366-370)
3. Expand the introduction slightly to incorporate more recent theoretical advancements in cognitive and educational psychology.
Revised. You can find it on page 2 (lines 42-74) in the file I uploaded to the system
Kind regards.

Reviewer 2 Report
Comments and Suggestions for Authors
-
Author Response
thank you
Reviewer 3 Report
Comments and Suggestions for Authors
The authors did a good job with revision, The readability of the paper increased and it is now clear enough. This version can be accepted for publication. Congratulations!
Author Response
thank you